# IMPROVING LANGUAGE AGENTS THROUGH BREW

## ABSTRACT

Large Language Model (LLM)-based agents are increasingly applied to tasks requiring structured reasoning, tool use, and environmental adaptation, such as data manipulation, multistep planning, and computer-use automation. However, despite their versatility, current training paradigms for model weight optimization methods, like PPO and GRPO, remain relatively impractical with their high computational overhead for rollout convergence. In addition, the resulting agent policies are difficult to interpret, adapt, or incrementally improve. To address this, we investigate creating and refining structured memory of experiential learning of an agent from its environment as an alternative route to agent optimization. We introduce **BREW** (Bootstrapping expeRientially-learned Environmental knoWledge), a framework for agent optimization for downstream tasks via KB construction and refinement. In our formulation, we introduce an effective method for partitioning agent memory for more efficient retrieval and refinement. BREW uses task graders and behavior rubrics to learn insights while leveraging state-space search for ensuring robustness from the noise and non-specificity in natural language. Empirical results on real world, domain-grounded benchmarks – OSWorld and $\tau^2$Bench – show BREW achieves $10 - 20\%$ improvement in task precision, $10 - 15\%$ reduction in API/-tool calls leading to faster execution time, all while maintaining computational efficiency on par with base models. Unlike prior work where memory is treated as static context, we establish the KB as a modular and controllable substrate for agent optimization – an explicit lever for shaping behavior in a transparent, interpretable, and extensible manner.

## 1 INTRODUCTION

Large Language Model (LLM) based agents are rapidly being deployed for structured reasoning, tool use, and autonomous interaction in real-world environments (Li, 2025). From computer-use and spreadsheet automation to software engineering pipelines, these agents drive tasks such as multi-step planning, data manipulation, and adaptive workflows (Qin et al., 2025; Jimenez et al., 2024; Yang et al., 2024; Anthropic, 2024; OpenAI, 2025). For example, a language agent might help automate a multi-step workflow like collecting data from different sources, cleaning or validating it, and then uploading it onto a dedicated server, all while adjusting its plan if the format or structure of the data changes unexpectedly (Yang et al., 2023; Zhou et al., 2024; Shinn et al., 2023; Bajpai et al., 2024). Yet, despite these successes, top-performing agents generally score underwhelmingly on challenging real-world benchmarks—well behind human experts (Yao et al., 2024; Barres et al., 2025a; Xie et al., 2024; Ma et al., 2024). As an example, consider the following scenario:

---

**Case Study on Computer Use Agents**

A computer-use agent in an Ubuntu environment tasked with automating software installation across multiple sessions.

In its first encounter, it struggles through a 47-step process: opening the wrong package manager, executing redundant dependency checks, and making 23 API calls to complete what could be a 6-step workflow.

When presented with a similar installation task in the next session, the agent repeats the same inefficient exploration — *as if encountering the problem for the first time.*

A human user, by contrast, would likely have a recollection from internalized memory of the optimal sequence after the first attempt, recognizing the environmental patterns."

---

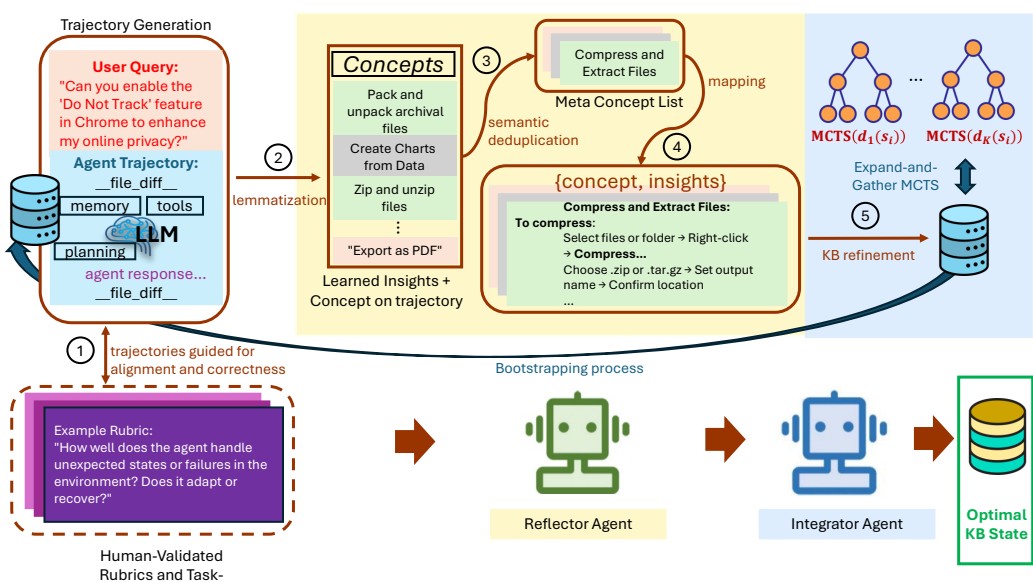

Figure 1: BREW architecture overview using examples from the OSWorld dataset. Step 1 indicates the trajectory generation process with agent alignment to human-validated rubrics and correctness using task-specific grader. Steps 2–4 indicate the Reflector Agent, which learns key concepts and corresponding insights from trajectories. Step 5 indicates the Integrator Agent, which integrates knowledge from the Reflector Agent to bootstrap the KB. We introduce Expand-and-Gather MCTS for finding the best KB configuration by a reward-guided search.

This scenario illustrates a fundamental limitation of current language agents: despite their impressive capabilities in reasoning and tool use, they lack the ability to accumulate and apply experiential knowledge across task sessions. Each interaction begins from a blank slate, forcing agents to repeatedly explore the same action spaces and rediscover the same solutions (Erdogan et al., 2025). Real-world tasks like long horizon multi-stage automation demand more than just "reactive" (Yao et al., 2023) tool loops. They require persistent & interpretable learnings from past experiences - what works, what fails and why.

To close this gap, recent work has explored learning agent behavior using model weight optimization (Schulman et al., 2017; Rafailov et al., 2024; Shao et al., 2024), where agents are trained to maximize success across a wide variety of tool-use episodes. However, while conceptually sound, this suffers from practical limitations. First, it requires expansive exploration over large rollout spaces to converge, especially in domains where tasks are diverse, goals are sparsely defined, and intermediate feedback is noisy or delayed. Second, the resulting policies are often opaque—difficult to interpret, revise, or debug—limiting their real-world deployability. Finally, these policies are tightly coupled to the task distributions they were trained on, making it difficult to adapt or incrementally improve them when downstream requirements shift.

In contrast, others have explored learning of knowledge onto a memory module that remains attached to an agent. These existing memory-augmented agents can be broadly classified into either ones which (i) store only transient trajectory contexts that vanish between episodes like Mem0 (Chhikara et al., 2025; Xu et al., 2025b), or (ii) embed high-level notes directly in the prompt such as MetaReflection (Gupta et al., 2024b) and GEPA(Agrawal et al., 2025). While the latter often do not retain actionable details for future simple tasks, neither of these approach supports modular updates, fine-grained retrieval, or transparent inspection of what the agent "knows." (Xu et al., 2025a).

Leveraging learnings from both camps, we introduce BREW (**B**ootstrapping expe**r**ientially-learned **e**nvironmental kno**w**ledge), a framework that incrementally constructs and refines a knowledge base (KB) a structured collection of concept-level documents in natural language, directly from an agent's

past interactions. This KB then serves as a persistent memory for the agent to retrieve knowledge in future executions to improve precision and efficiency outcomes. Our key contributions are–

- *Novel experience-driven KB construction.* We propose a technique for leveraging agent's past interaction trajectories to generate uniquely-partitioned concept-level KB documents. This process is guided by rubrics and task-specific graders which ensures that memories are both semantically aligned with task objectives and human-interpretable.

- *State-space search for memory optimization.* We formalize the selection and update of KB entries as a state search problem and introduce an efficient reward-guided learning scheme, Expand-and-Gather Monte Carlo Tree Search (EG-MCTS), that learns to prioritize the most impactful memories for robust, multi-step reasoning.

- *State-of-the-art results.* On domain-grounded benchmarks including OSWorld and $\tau^2$Bench, BREW achieves significant gains of in the range of $10 - 20\%$ towards task precision as well as $10 - 15\%$ fewer steps leading to faster execution, while maintaining memory and compute costs comparable to base LLMs.

## 2 RELATED WORKS

**Agent Learning from Demonstrations**  Recent work has leveraged LLMs to isolate reusable skills through interactive decomposition (Hashemzadeh et al., 2024), synthesizing executable domain specific functional abstractions (Khan et al., 2025) or by learning in-prompt memory (Gupta et al., 2024b). These approaches focus on structured skill extraction from LLM-guided interactions, yet remain reliant on static decomposition or offline synthesis. In contrast, BREW dynamically constructs and refines an experiential memory learning necessary semantic fragments via rollout generated insights and structured knowledge-base search (MCTS) to support long-horizon, memory augmented planning. Besides unlike prompt optimization based techniqes (Agrawal et al., 2025; Gupta et al., 2024b), BREW represents learning as retrievel agent memory knowledge bases, providing extensibility to the memory.

**Agentic Memory**  The concept of providing agents with controllable memory has a rich history. (Littman, 1993). Memory mechanisms are attracting more and more attention lately (Packer et al., 2024; Wang et al., 2025; Xu et al., 2025a; Chhikara et al., 2025; Xu et al., 2025c; Hu et al., 2025). These works focus towards storing relevant context in a structured format like graph or a tree so as to RAG over it. While these techniques work well for sub-domains they are designed for, they fail to generalize (Hu et al., 2025). In contrast, BREW uses a reward driven state exploration to select the memory states making it more robust to ambiguous queries and especially useful in multi-turn settings.

**State Based Explorations**  State-space search has been extensively used for exploration based learning (Silver et al., 2016; Liu et al., 2025). With he advent of prompt-tuned LLM systems, state space techniques are being actively explored in the community for text-based optimization (Gupta et al., 2024a; Wang et al., 2023; Novikov et al., 2025). Notably, our technique builds upon this work and generalizes it to general purpose Agent Memory Learning.

## 3 BREW: ARCHITECTURE

This section describes our proposed **B**ootstrapping expe**R**ientially-learned **E**nvironmental kno**W**ledge technique, BREW, which constructs and iteratively refines a Memory KB using trajectory insights guided by human-validated general-purpose agent behavior metrics, task-specific evaluation, and latent insight generation. We decompose the problem of learning the optimal KB by partitioning memory as local documents associated with semantic concepts, and solve the KB learning problem by our novel Expand-and-Gather Monte Carlo Tree Search (EG-MCTS) algorithm. Figure 1 provides an architecture overview of BREW, and Algorithm 1 describes the pseudocode.

### 3.1 TRAJECTORY GENERATION

Given the training dataset, we generate full-length trajectories, hereby referred to as rollouts, for each query using an LLM-powered agent conditioned on its associated KB. At initialization, the KB is

empty, and we generate rollouts with an *empty* KB. Each rollout is evaluated using a correctness grader, which assigns a binary success label and an LLM based qualitative assessment against a set of human-validated general-purpose agent behavior rubrics (Biyani et al., 2024) (Step 1 in Figure 1).

## 3.2 REFLECTOR AND INTEGRATOR AGENTS

**Reflector Agent:** `ReflAgent` takes as input a rollout with its rubric and correctness labels, and outputs sentence-level insights with mapped concepts:

$$\{concepts, insights\} = \texttt{ReflAgent}(\{rollout, eval\}). \tag{1}$$

Examples of concept–insight pairs appear in Step 2 of Figure 1.

**Concept Deduplication:** Concept–insight pairs are annotated independently per rollout, often producing overlapping or paraphrased concepts. We address this via semantic clustering (Steps 3–4, Figure 1; Algorithm 1, line 3): contextual embeddings for each concept are generated using an LLM, clustered, and each insight is mapped to its cluster representative. Details appear in Algorithms 2 and 3 in Appendix A.

**Integrator Agent:** `IntegAgent` incrementally builds and refines KB documents $\{d(s_i)\} \in \mathcal{D}(s_i)$ during environment interaction. Instead of a centralized memory, the KB is partitioned into local documents, each tied to a meta concept. This design enables (1) efficient, context-specific retrieval; (2) modular updates with minimal interference; and (3) natural alignment with task semantics, as deduplicated meta concepts capture meaningful behavioral abstractions. Unlike prior work assuming flat memory or dialogue histories, this structure is well-suited for long-horizon, procedural tasks where behaviors cluster around discrete skills.

The KB is dynamically populated: concepts central to the dataset receive more updates, shaping memory around frequent behaviors. At each state, for meta concept $k$, `IntegAgent` updates its document $d_k$ via

$$d_k(s_{i+1}) \leftarrow \texttt{IntegAgent}(k, insights_k, d_k(s_i)). \tag{2}$$

To reduce LLM variance and improve consistency, we use the Expand-and-Gather MCTS (EG-MCTS) method (Figure 2).

Formally, the KB at state $s_i$ is the union of all concept-localized documents:

$$\mathcal{D}(s_i) = \bigcup_{k \in \mathcal{K}} \{d_k(s_i)\}, \tag{3}$$

where $\mathcal{K}$ is the set of all meta concepts and $d_k(s_i)$ is the document for concept $k$ at state $s_i$.

## 3.3 EXPAND-AND-GATHER MCTS FOR OPTIMAL KB SEARCH

We start by creating a set of meta-concepts after deduplicating concepts extracted by `ReflAgent` using the first set of trajectory rollouts. We freeze this meta-concept set $\mathcal{K}$, and use it to initialize a KB with an empty document per concept $k \in \mathcal{K}$.

We model the problem of finding the optimal KB $\mathcal{D}^*$ as a search problem in the *state* space of all possible KBs $\mathcal{D}$. To simplify this state search, we model KB $\mathcal{D}$ as a collection of concept level documents. This modeling allows us to break down the larger search space into a collection of simpler document level search problems for each concept $k$ to find the optimal document $d_k^*$. We then construct the optimal KB $\mathcal{D}^*$ by combining all optimal documents $d_k^*$ for each concept $k$ as follows:

$$\mathcal{D}^* = \bigcup_{\forall k} \{d_k^*\} \tag{4}$$

Notably, even though we are modeling document level search as independent optimization problems, each document in the KB is *not* independent of the others. For example, an agent can retrieve any document in the KB during inference and this retrieval making it hard to assess the impact of changing a document in isolation. To solve this we propose Expand-and-Gather MCTS (EG-MCTS), which enables searching these disjoint state spaces concurrently using parallel MCTS explorations that are synced after each iteration. To achieve this we perform node expansions in the respective search spaces independently but condition reward calculation and insight generation on a running optimum KB state. Each iteration of EG-MCTS can be broken down two phases:

---

**Algorithm 1** BREW: Bootstrapping Experientially-learned Environmental Knowledge

---

**Require:** Training samples $\mathcal{Q}_{\text{train}}$, eval samples $\mathcal{Q}_{\text{eval}}$, rubrics, iterations $M$, candidates per expansion $h$

**Ensure:** Optimized KB $\mathcal{D}^*$

    **Initialization**

1: $\mathcal{D}_0 \leftarrow \varnothing$
2: $\mathcal{B} \leftarrow \text{GENERATEINSIGHTS}(\mathcal{Q}_{\text{train}}, \mathcal{D}_0, \text{rubrics})$
3: $\mathcal{K} \leftarrow \text{DEDUPLICATECONCEPTS}(\mathcal{B})$           ▷ Initial concept set
4: **for** each $k \in \mathcal{K}$ **do**
5:      $d_k^0 \leftarrow \text{INTEGAGENT}(k, \mathcal{I}_k, \varnothing)$
6:      Initialize $\text{tree}_k$ with root node $d_k^0$
7: **end for**
8: $\mathcal{D}_{\text{current}} \leftarrow \bigcup_{k \in \mathcal{K}}\{d_k^0\}$           ▷ Initial KB
    **EG-MCTS Optimization**
9: **for** $t = 1$ to $M$ **do**           ▷ Parallel expansion across concepts
10:      **for** each $k \in \mathcal{K}$ **do**
11:          $s_k \leftarrow \text{SELECTBESTNODE}(\text{tree}_k)$           ▷ UCT selection
12:          $\mathcal{D}_{\text{best}} \leftarrow \bigcup_{k' \in \mathcal{K}}\{d_{k'}^{\text{best}}\}$           ▷ Current best docs
13:          $\text{EXPANDNODE}(s_k, k, h, \mathcal{D}_{\text{current}}, \mathcal{D}_{\text{best}}, \text{tree}_k)$
14:      **end for**           ▷ Update current best documents
15:      **for** each $k \in \mathcal{K}$ **do**
16:          $d_k^{\text{best}} \leftarrow$ best document in $\text{tree}_k$
17:      **end for**
18:      $\mathcal{D}_{\text{current}} \leftarrow \bigcup_{k \in \mathcal{K}}\{d_k^{\text{best}}\}$
19: **end for**
20: **return** $\mathcal{D}_{\text{current}}$

**Time Complexity:** $O(|\mathcal{Q}_{\text{train}}| \cdot T_{\text{LLM}} + M \cdot |\mathcal{K}| \cdot h \cdot T_{\text{agent}})$

---

**Expand Phase:** During this stage, for each search tree, we pick the *best* state $s^*$ and expand it concurrently. To perform this expansion the KB $\mathcal{D}(s^*)$ is constructed by including the *current document* $d_k(s^*)$ and the *best (oracle) documents* $\{d_i^*\}_{i \neq t}$ for all other positions. Thus, the KB at iteration $t$, $0 \leq t \leq E$ is defined as:

$$\mathcal{D}_t = d_t \cup d_{i:i \neq t}^* \tag{5}$$

We use this KB $\mathcal{D}(s_i)$ to generate trajectory rollouts which are consumed by the `ReflAgent` to generate insights. We then use the `IntegAgent` to generate various updated variants of $d_k^*$ e.g., $d_k(s_i), ..., d_k(s_j)$, where $0 \leq i \leq E$ and $0 \leq j \leq E$. We then estimate a reward $R$ for each of these newly generate states and update rewards of parent states using backpropagation.

**Gather Phase:** During this stage, the current best states from each document's MCTS tree are *gathered* together and distributed to every MCTS tree for reward calculation. This is important to (i) Estimate rewards for each expanded state, & (ii) Generate new insights for further node expansion.

### 3.4 REWARD-GUIDED OPTIMIZATION

This section describes BREW's joint reward and loss optimization for learning an optimal KB.

**Reward Objective:** Each document state is rewarded based on two complementary criteria: *(i)* how well the current document contributes to **accurate downstream reasoning**, and *(ii)* how **retrievable** it is in the context of a growing KB. Formally, the total reward at time step $t$ is defined as:

$$R_t = \lambda_{\text{corr}} \cdot R_t^{\text{corr}} + \lambda_{\text{ret}} \cdot R_t^{\text{ret}} \tag{6}$$

where $R_t^{\text{corr}}$ is the **correctness reward**, $R_t^{\text{ret}}$ is the **retrieval reward**, and $\lambda_{\text{corr}}, \lambda_{\text{ret}} \in [0, 1]$ are scalar weights with $\lambda_{\text{corr}} + \lambda_{\text{ret}} = 1$.

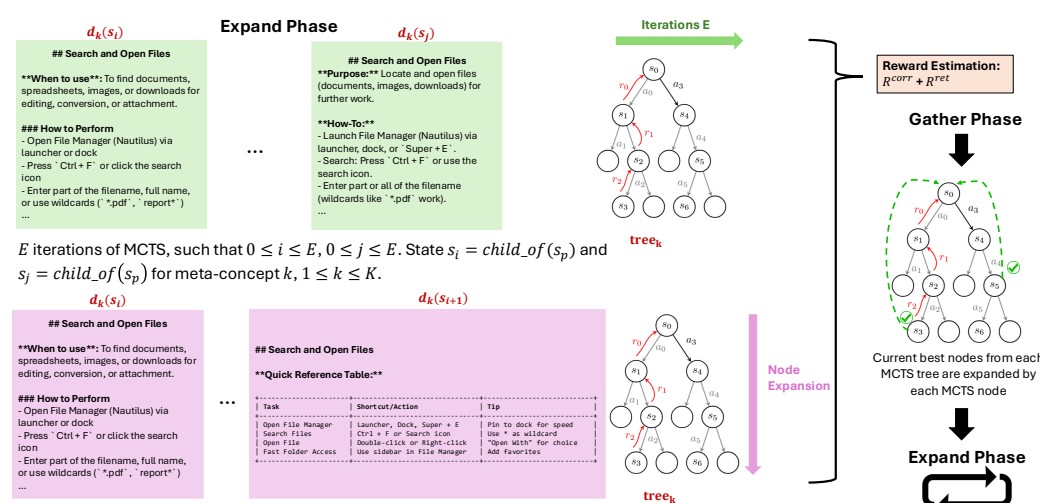

Node expansion at state $s_i$ for meta-concept $k$, $1 \le k \le K$.

Figure 2: Illustration of BREW's KB optimization process using Expand-and-Gather MCTS with OSWorld examples. In the **Expand Phase**, for each document $k$, we sample the best node from tree$_k$ using UCT and perfrom node expansion. Node rewards are estimated based on correctness and retrievability. In the **Gather Phase**, the current best nodes from each tree are gathered at each node. The process is repeated for the next iteration of KB refinement.

**Correctness Reward:**  The correctness reward $R_t^{\text{corr}}$ evaluates the accuracy of the agent's output over a held-out query set $\mathcal{Q}$, when reasoning over the current KB $\mathcal{D}_t$. It is defined as:

$$R^{\text{corr}}(d_t|\mathcal{D}_t) = \frac{1}{|\mathcal{Q}|} \sum_{q \in \mathcal{Q}} \text{Eval}_{\text{task}}(q, \texttt{agent} \oplus \mathcal{D}_t) \tag{7}$$

where $\text{Eval}_{\text{task}}$ is a task-specific evaluation function (e.g., question-answering accuracy, entailment correctness), and $\texttt{agent} \oplus \mathcal{D}_t$ denotes the agent acting over the hybrid KB.

**Retrieval Reward:**  The retrieval reward $R_t^{\text{ret}}$ measures how effectively the current document $d_t$ can be retrieved from the current KB $\mathcal{D}_t$. For a held-out query set $\mathcal{Q}$, it is computed using the mean reciprocal rank (MRR):

$$R^{\text{ret}}(d_t|\mathcal{D}_t) = \frac{1}{|\mathcal{Q}|} \sum_{q \in \mathcal{Q}} \text{MRR}_q(d_t, \mathcal{D}_t) \tag{8}$$

This encourages documents that are not only helpful in reasoning but also easily retrievable over $\mathcal{D}_t$.

## 4  EXPERIMENTAL SETUP

**Datasets**  We evaluate BREW on three diverse benchmarks testing different aspects of interactive agent capabilities: OSWORLD for computer-use automation (Xie et al., 2024), $\tau^2$-Bench for tool use (Barres et al., 2025b), and SPREADSHEETBENCH for data manipulation (Ma et al., 2024).

1. **OSWorld:** This benchmark tests multimodal agents on real-world computer tasks across 10 applications. We use *GTA1-7B*, a state-of-the-art computer-use agents with BREW. Tasks are evaluated using 134 custom scripts that verify final application states.

2. **$\tau^2$-Bench:** This benchmark evaluates conversational agents on multi-turn tool-use scenarios across *Telecom*, *Retail*, and *Airline* domains. We test `o4-mini`-based tool-calling agent, constructing BREW KBs for every domain.

3. **SpreadsheetBench:** This benchmark evaluates agents on real-world spreadsheet manipulation, spanning both cell-level and sheet-level tasks. It contains 912 authentic user instructions paired

with 2,729 test cases (3̃ per instruction), sourced from Excel forums and blogs. Spreadsheets include diverse formats with multi-table sheets (35.7%) and non-standard tables (42.7%). We test `o4-mini` using a Python tool-calling agent, and enhance it with by adding an embedding based Retrieval over the BREW KB generated over a small held-out train set of 30 samples.

**Baselines** We compare BREW against two widely used experiential memory approaches, *Cognee*[1] (Markovic et al., 2025) and *Agent-Mem* (Xu et al., 2025c), both of which serve as established baselines for AI memory evaluation. Cognee is an open-source AI memory engine that employs a graph-plus-vector memory architecture through an Extract–Connect–Learn pipeline, enabling agents to construct cross-document and cross-context connections entirely from previously available trajectories. In contrast, Agent-Mem provides a scalable memory layer for dynamically extracting and retrieving information from conversational data, with enhanced variants incorporating graph-based memory representations. While Cognee primarily emphasizes cross-document relational reasoning, Agent-Mem focuses on scalable personalization for conversational agents.

**Other Experimental Configs:** For all experiments, we use `GPT-4.1-2025-04-14` as the base LLM with expansion width $e = 3$, max depth $k = 3$, and balanced reward weights $\lambda_{corr} = \lambda_{ret} = 0.5$. During MCTS node selection, we use the UCT (Kocsis and Szepesvári, 2006) for balancing exploration and exploitation Full experimental details are provided in the Appendix.

## 5  ANALYSIS & DISCUSSION

In this section, we present findings from our evaluation of BREW. For more details on qualitative insights and discussion you may refer to the supplementary material.

### 5.1  VARIATIONS ACROSS STATE SEARCH STRATEGY

BREW performs a search across possible KB states using MCTS. We compare different state search strategies to determine the relative trade-offs:

1. *Iterative Refinement*: In this strategy we generate one version of each document to generate an initial KB, followed by a round of evaluations. We then use the aggregator agent to refine the documents over the newly learned insights. We repeat this step multiple times up to a maximum number of refinements. Note that in contrast to MCTS, in this strategy we *do not* perform node expansions and rather explore a path in the search tree.

2. *Greedy Search*: In this strategy we greedily pick the best state during each node expansion and only explore the sub-tree within it. This is in contrast to MCTS where, we explore different states using the UCT algorithm that balances exploration and exploitation.

Table 1 presents how MCTS achieves consistent performance gains across all benchmarks. These represent 1-5% improvements over alternative search strategies across tasks. Iterative refinement's poor performance reveals core limitations in the integrator agent feedback incorporation- which can be attributed to inherent stochasticity in LLMs. This makes state exploration especially important for textual optimization tasks like ours. We present a detailed analysis on how varying MCTS parameters result in different final states in appendix.

### 5.2  TRENDS ACROSS SUB-TASKS

**BREW learns recipes from sub-trajectories in OSWorld.** Figure 3 shows that BREW (BREW) improves success rates in 5 out of 10 OSWorld categories, achieving absolute gains of 4–16% while maintaining performance parity in the remaining categories (Chrome, Gimp, LibreOffice Calc, LibreOffice Impress, OS). The largest improvements appear in text-processing applications (LibreOffice Writer: $14\% \rightarrow 24\%$, Thunderbird: $38\% \rightarrow 54\%$) and multimedia tools (VLC: $20\% \rightarrow 27\%$), with moderate gains in multi-application and development environments. Even in settings with limited improvements in task correctness, BREW consistently reduces execution length by 14–23 steps, highlighting more efficient planning. This pattern suggests that BREW's

---

[1]github.com/topoteretes/cognee

| Method | OSWorld GTA1-7B | $\tau^2$ **Bench** o4-mini | **SpreadsheetBench** o4-mini |
|---|---|---|---|
| Baseline | 44.20 | 56.63 | 44.30 |
| Cognee | 46.70 | 57.71 | 42.10 |
| Agent-Mem | 43.83 | 52.69 | 42.00 |
| BREW -Iterative | 46.13 | 57.34 | 42.98 |
| BREW -Greedy | 45.55 | 59.14 | 45.94 |
| BREW -MCTS | **47.56** | **59.14** | **46.80** |

Table 1: Comparison of models under different evaluation setups, including Baseline model and BREW augmented model. We report task success rate for OSWorld, ratio of independent tasks that succeeded for $\tau^2$ Bench, and the 1st test case pass rate for SpreadsheetBench.

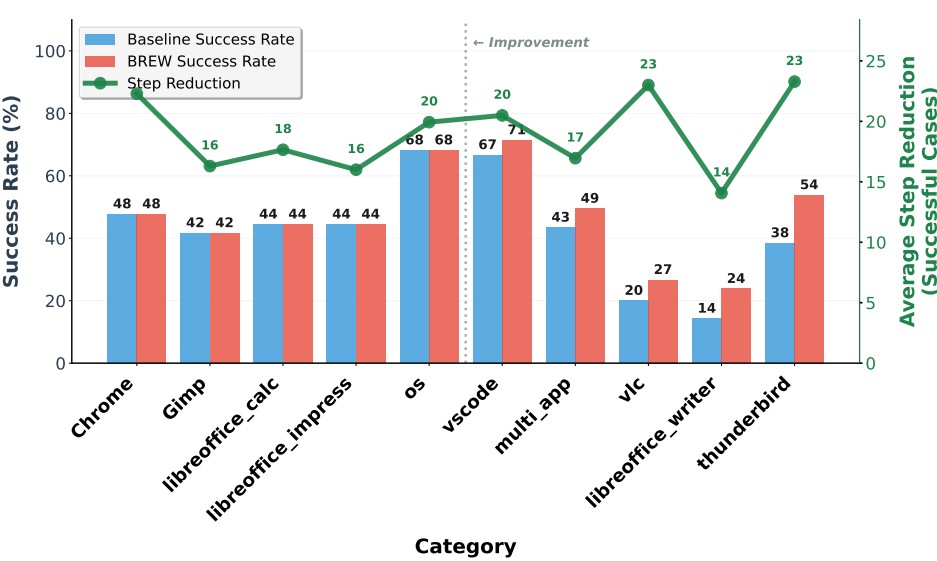

Figure 3: The bar plot represents the category-wise success rate over various tasks in the OSWorld dataset over the GTA1-agent, whereas the line plot demonstrates the reduction in the number of steps for the successful cases. Note that even in scenarios where the KB doesn't help increase the success rate, it significantly reduces the number of steps needed to succeed.

architectural enhancements are particularly effective for tasks requiring complex sequential reasoning and inter-application coordination, while preserving baseline robustness in domains constrained by intrinsic task complexity.

A qualitative analysis of the knowledge bases (KBs) constructed by BREW further supports this finding. We observe that BREW captures and represents sub-trajectory characteristics in *natural language*, including application shortcuts, standard operating procedures, and strategies for localizing UI elements. Since many UI tasks share common sub-trajectories, this representation facilitates knowledge transfer across tasks within the same application. Moreover, BREW substantially reduces reliance on granular UI interactions: while the baseline GTA1 model executes approximately 19,000 clicks and 17,821 keyboard actions, BREW significantly decreases this interaction complexity.

**BREW learns aggressive resolution strategies for $\tau^2 - Bench$** To evaluate robustness of BREW, we analyzed the distribution of failure modes across the $\tau^2$–retail dataset, focusing on four key error categories: *Wrong Argument*, *Wrong Info*, *Wrong Decision*, and *Partially Resolve*. Figure 4 presents a comparative chart for the baseline, BREW, Cognee and Agent-Mem.

Overall, BREW demonstrated consistent improvements across most error types compared to the baseline and competing approaches. Specifically, BREW showed a **notable reduction in "Wrong**

**Argument" and "Wrong Decision" errors**, indicating that it was better at capturing logical dependencies in retail dialogues and making accurate decisions. Interestingly, *Partially Resolve* errors were slightly higher for BREW than for Cognee, likely because BREW attempted more aggressive resolution strategies that occasionally failed to fully satisfy user queries. Cognee appears to capture *richer factual details* given its relatively lower *Wrong Info* errors, whereas Agent-Mem excels in *tracking conversation state* and *decision accuracy*, as reflected in its reduced *Wrong Decision* failures.

**BREW learns domain specific strategies for SpreadSheetBench** BREW shows consistent improvements over the Baseline for SpreadSheetbench, powered by domain specific insights learnt in the KB. Specifically, we observed that most improvements came with a more precise placement of formulas, in 90% of the cases brew showed improvements over the baseline, the difference was the correct placement of code. This is followed by double checking the results before submitting at 85%, and using the filter formula correctly at 65% of the cases.

**Improvements in Task Efficiency** We observe that overall, BREW enables agents to come to a correct response in fewer steps compared to baseline.

*OSworld.* Figure 3 demonstrates that BREW enables GTA1 to complete tasks more efficiently. Compared to the baseline GTA1 model's average of ∼75 steps, the BREW-augmented model completes tasks 14% faster with an average of ∼64 steps. Analyzing performance by outcome reveals that while step counts remain unchanged for failed cases, successful completions show a substantial 39% (rel.) reduction in execution steps, indicating improved planning efficiency for achievable tasks.

$\tau^2$*Bench.* Similarly, BREW reduces average conversation turns from 29.47 to 28.43 (-3.5%), while maintaining consistent step reductions across categories. Step reductions average 1.7 steps for Retail and Telecom, but 3.1 steps for Airline, indicating greater efficiency gains in complex domains. Qualitative analysis seconds these numbers showing how knowledge base integration enables more direct task completion paths and improved planning quality, though multi-turn interactions remain necessary for complex sub-tasks.

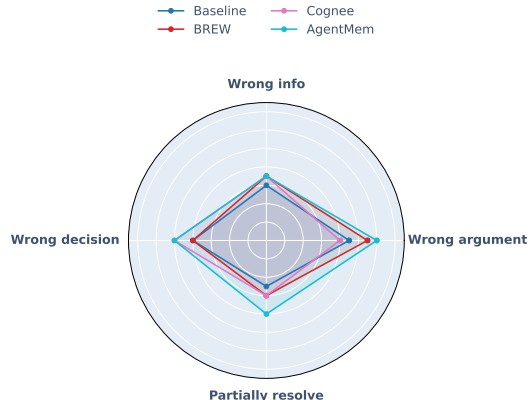

Figure 4: Distribution of errors in $\tau^2$ Bench Retail

*SpreadsheetBench.* While we observe a slight increase in the number of turns across the entire benchmark suite ($4.5 \rightarrow 5.4$) in the case of the baseline versus BREW, an interesting pattern emerges in more than $82\%$ of the cases the baseline and the BREW appended agent performs similarly with similar turn consumption. BREW leads to an improvement in 12% of the cases where the KB is able to address gaps in the baseline technique to enable the agent to go exploring further leading to positive outcomes with an average of 1 step increase in the interactions.

## 6 CONCLUSIONS

In this work, we explored an alternative approach to agent optimization by focusing on experiential knowledge retention rather than direct model fine-tuning. We introduced BREW, a framework that aims to construct and refine a structured, interpretable knowledge base from past agent interactions. By decomposing agent memory into concept-level documents and applying a state-search optimization strategy, BREW provides a modular and transparent substrate for memory formation. Our evaluations across OSWorld and $\tau^2$Bench benchmarks suggest that such structured memory can support measurable improvements in task success and efficiency, while maintaining manageable computational costs. Although the observed gains are promising, we recognize that BREW's effectiveness is influenced by the quality and coverage of its training data. Future work could explore more adaptive and domain-general memory refinement techniques, as well as tighter integrations with ongoing agent planning. Ultimately, we hope this study encourages further investigation into more interpretable, memory-driven approaches to language agent development—especially in real-world environments where long-term consistency and adaptability are essential.

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

# A Appendix

## A.1 Details of the BREWAlgorithm

We provide pseudocode for the core components of BREW, aligning with the stages introduced in Section 3. Each algorithm plays a distinct role in constructing, organizing, or refining the knowledge base over iterative interactions. GENERATEINSIGHTS (Alg. 2) produces concept-aligned insights from annotated rollouts using `ReflAgent`. DEDUPLICATECONCEPTS (Alg. 3) clusters semantically overlapping concepts into a compact meta-concept set. INTEGAGENT incrementally builds and updates per-concept documents using newly generated insights. Finally, EXPANDNODE (Alg. 4) performs MCTS-guided expansions to explore improved document variants, while EVALUATE (Alg. 5) scores candidate KB states using correctness and retrieval-based rewards.

We specify the `IntegAgent` prompt below:

**BREW Integrator Prompt**

```
# Enhanced Documentation Editor Prompt

You are a meticulous documentation-level editor specializing in
    comprehensive technical reference materials. You will be given a
    list of topic nodes, each containing structured information that
    must be preserved and enhanced with maximum detail retention.

## Input Structure Analysis
Each node contains:
- **Title**: The primary topic identifier
- **Context**: Background information and conceptual foundation
- **How to Use**: Step-by-step instructions, commands, flags,
    parameters, and implementation details
- **When to Use**: Specific scenarios, conditions, and decision
    criteria
- **Best Practices**: Expert recommendations, optimization techniques,
     and common pitfalls to avoid

## Detailed Processing Requirements

### 1. Information Preservation (Zero Loss Policy)
- **Preserve every technical detail**: All command-line flags,
    parameter values, configuration options, file paths, URLs, version
     numbers, and exact syntax
- **Maintain all examples**: Keep every code snippet, sample input/
    output, file names, directory structures, and command sequences
    exactly as provided
- **Retain contextual nuances**: Preserve qualifying language like "
    typically," "usually," "in most cases," "when available," and
    conditional statements
- **Keep quantitative data**: Preserve all numbers, measurements,
    timeframes, limits, thresholds, and statistical information
- **Maintain cross-references**: Keep all mentions of related tools,
    dependencies, prerequisites, and interconnected concepts

### 2. Enhanced Detail Extraction
- **Expand abbreviations**: When encountering shortened forms, expand
    them naturally while preserving the original
- **Surface implicit knowledge**: Make obvious assumptions explicit (e
    .g., "this requires root permissions," "assumes default
    configuration")
- **Clarify relationships**: Explicitly describe how different
    components, options, or steps relate to each other
- **Highlight edge cases**: Emphasize special conditions, exceptions,
    or unusual scenarios mentioned in the source
- **Elaborate on consequences**: When the source mentions outcomes,
    expand on both success and failure scenarios
```

### 3. Prose Transformation Guidelines
- **Bullet integration**: Transform each bullet point into 1-3 complete sentences that naturally flow together
- **Technical precision**: Use precise technical vocabulary while maintaining readability
- **Logical flow**: Organize information within each section to follow a logical sequence (setup →execution →verification)
- **Contextual embedding**: Weave code snippets and technical terms seamlessly into narrative sentences
- **Comprehensive coverage**: Ensure every sub-bullet, nested item, and parenthetical note becomes part of the prose

### 4. Structural Requirements
- **Heading hierarchy**: Use `# Title` for each node's main heading
- **Section order**: Maintain Context →How to Use →When to Use →Best Practices sequence
- **Paragraph organization**: Create substantial paragraphs (3-6 sentences) rather than brief statements
- **Transition quality**: Craft smooth bridges between sections and between different nodes
- **Code formatting**: Preserve all inline code with backticks and maintain proper formatting for code blocks

### 5. Quality Assurance Checklist
Before finalizing, verify:
- [ ] Every piece of source information appears in the output
- [ ] All technical specifications, parameters, and examples are intact
- [ ] Code snippets maintain their exact syntax and formatting
- [ ] Prose flows naturally without choppy or fragmented sentences
- [ ] Each section provides comprehensive coverage of its topic area
- [ ] Cross-references and dependencies are clearly explained
- [ ] No section labels or formatting artifacts remain in the prose

## Output Specifications
Generate a single, cohesive markdown document that reads as authoritative technical documentation. The result should be comprehensive enough that a reader could successfully implement the described tools or techniques using only the information provided, without referring back to the original nodes.

---

**Input Nodes:**
<NODES>
{node_list}
</NODES>

---

Now, produce the aggregated markdown reference sheet with maximum detail preservation and enhanced clarity.

**Algorithm 2** GenerateInsights: Extract behavioral insights from trajectories

---

**Require:** Queries $\mathcal{Q}$, KB $\mathcal{D}$, rubrics
**Ensure:** Concept-insight pairs $\mathcal{B}$

1: $\mathcal{B} \leftarrow \varnothing$
2: **for** each query $q \in \mathcal{Q}$ **do**
3: $\quad \tau \leftarrow \text{LLM}(q, \mathcal{D})$         ▷ Generate trajectory
4: $\quad \text{label} \leftarrow \text{GRADE}(\tau)$         ▷ Success/failure
5: $\quad (c, i) \leftarrow \text{REFLAGENT}(\tau, \text{rubrics}, \text{label})$
6: $\quad \mathcal{B} \leftarrow \mathcal{B} \cup \{(c, i, q)\}$         ▷ Store with source query
7: **end for**
8: **return** $\mathcal{B}$

---

**Algorithm 3** DeduplicateConcepts: Cluster similar concepts and map queries

---

**Require:** Concept-insight-query triples $\mathcal{B}$
**Ensure:** Meta-concepts $\mathcal{K}$ with mapped queries and insights

1: Extract all concepts from $\mathcal{B}$
2: Embed and cluster concepts by similarity
3: $\mathcal{K} \leftarrow$ cluster representatives
4: **for** each $k \in \mathcal{K}$ **do**
5: $\quad \mathcal{Q}_k^{\text{train}} \leftarrow \{\text{training queries that contributed insights to } k\}$
6: $\quad \mathcal{Q}_k^{\text{eval}} \leftarrow \{\text{held-out queries relevant to } k\}$
7: $\quad \mathcal{I}_k \leftarrow \{\text{all insights mapped to concept } k\}$
8: **end for**
9: **return** $\mathcal{K}$ with associated queries and insights

---

**Algorithm 4** ExpandNode: Generate and evaluate new document variants

---

**Require:** Node $s$, concept $k$, candidates $h$, current KB $\mathcal{D}_{\text{current}}$, best docs $\mathcal{D}_{\text{best}}$, tree
**Ensure:** Updated tree with new evaluated nodes

1:         ▷ Generate new insights from concept-relevant queries
2: $\mathcal{B}_{\text{new}} \leftarrow \varnothing$
3: **for** query $q \in \mathcal{Q}_k^{\text{train}}$ **do**
4: $\quad \tau \leftarrow \text{LLM}(q, \mathcal{D}_{\text{current}})$
5: $\quad (c, i) \leftarrow \text{ANNOTATE}(\tau, \text{rubrics}, \cdot)$
6: $\quad$ **if** $c$ maps to $k$ **then**
7: $\quad\quad \mathcal{B}_{\text{new}} \leftarrow \mathcal{B}_{\text{new}} \cup \{i\}$
8: $\quad$ **end if**
9: **end for**
10:         ▷ Generate and evaluate candidate documents
11: **for** $j = 1$ to $h$ **do**
12: $\quad d_{k,j} \leftarrow \text{INTEGAGENT}(k, \mathcal{I}_k \cup \mathcal{B}_{\text{new}}, d_k^s)$
13:         ▷ Evaluate using hybrid KB with best docs from other concepts
14: $\quad \mathcal{D}_{\text{hybrid}} \leftarrow \{d_{k,j}\} \cup \{d_{k'} \in \mathcal{D}_{\text{best}} : k' \neq k\}$
15: $\quad R_{k,j} \leftarrow \text{EVALUATE}(d_{k,j}, \mathcal{D}_{\text{hybrid}}, \mathcal{Q}_k^{\text{eval}})$
16:         ▷ Add to tree and backpropagate
17: $\quad$ Add $(d_{k,j}, R_{k,j})$ as child of $s$ in tree
18: $\quad$ Backpropagate $R_{k,j}$ from new node to root
19: **end for**

---

**Algorithm 5** Evaluate: Score document using held-out queries

---

**Require:** Document $d_k$, hybrid KB $\mathcal{D}_{\text{hybrid}}$, eval queries $\mathcal{Q}_k^{\text{eval}}$
**Ensure:** Reward score $R$
1: $R^{\text{corr}} \leftarrow 0$
2: $R^{\text{ret}} \leftarrow 0$
3: **for** each $q \in \mathcal{Q}_k^{\text{eval}}$ **do**
4: $\quad R^{\text{corr}} \leftarrow R^{\text{corr}} + \text{EVAL}(q, \text{agent} \oplus \mathcal{D}_{\text{hybrid}})$
5: $\quad R^{\text{ret}} \leftarrow R^{\text{ret}} + \text{MRR}(d_k, q, \mathcal{D}_{\text{hybrid}})$
6: **end for**
7: $R^{\text{corr}} \leftarrow \frac{R^{\text{corr}}}{|\mathcal{Q}_k^{\text{eval}}|}$
8: $R^{\text{ret}} \leftarrow \frac{R^{\text{ret}}}{|\mathcal{Q}_k^{\text{eval}}|}$
9: **return** $\lambda_{\text{corr}} \cdot R^{\text{corr}} + \lambda_{\text{ret}} \cdot R^{\text{ret}}$

---

## A.2 BREW CONFIGURATIONS

**Base LLM Configuration**  For all BREWalgorithm steps, we use the OpenAI GPT-4.1-2025-04-14 model as the underlying language model. To balance exploration and stability, we set the temperature to 0.7 for the `IntegAgent` component to encourage diversity in sampled completions, while all other calls use a temperature of 0.1 for deterministic behavior. The search process employs an expansion width of $e = 3$, a maximum search depth of $k = 3$, and a maximum of $n = 10$ iterations. Reward signals are weighted equally across correctness and retrieval relevance, with $\lambda_{corr} = \lambda_{ret} = 0.5$.

## A.3 BASELINE METHODS

We compare BREWagainst two common reasoning baselines. Step-Back Prompting encourages backward reasoning by guiding the model to work from the final task objective back to the initial actions. In-Context Learning augments the input prompt with successful trajectories from related tasks, enabling the model to benefit from relevant prior examples without additional fine-tuning.

## A.4 BENCHMARK SPECIFICATIONS

### A.4.1 OSWORLD: COMPUTER-USE AUTOMATION

**Dataset Overview**  OSWorld (Xie et al., 2024) comprises 369 real-world computer-use tasks spanning 10 distinct applications. The benchmark is divided into train and test sets, with the distribution of tasks across domains shown in Table 2.

**Agent Specifications**  The UI-Tars-7B variant is a 7B-parameter multimodal transformer fine-tuned for graphical user interface understanding. It operates over an action space of PyAutoGUI commands (e.g., click, type, and key presses). The agent integrates a retrieval module that queries a task-relevant knowledge base using the user-provided description, with the top three retrieved items added to the system prompt. Inputs to the model consist of a screenshot of the active UI paired with the natural language task description.

The GTA1-7B configuration adopts a two-agent architecture, consisting of a planner and a grounding module. The planner (GTA-1-7B) generates the high-level action sequence, while the grounding module (OpenAI O3) verifies and refines each action before execution. Knowledge retrieval is incorporated differently for each component: the planner performs a single retrieval at the start of execution, which is persisted in its prompt, whereas the grounding module performs dynamic retrievals at each verification step.

**Evaluation Protocol**  Evaluation uses 134 task-specific scripts designed for automated verification. Success criteria include file state checks (e.g., validating `.xlsx` or `.docx` outputs), UI element validation to confirm correct interaction, and process completion checks to ensure that the intended automation sequence was executed successfully.

### A.4.2 $\tau^2$-BENCH: INTERACTIVE TOOL USAGE

**Dataset Overview**  $\tau^2$-Bench (Barres et al., 2025b) extends $\tau$-Bench by introducing bidirectional tool-calling capabilities. The dataset covers multiple service-oriented domains, with domain-level task distributions summarized in Table 3.

**Domain Characteristics**  The benchmark spans several domains with distinct task characteristics. The Telecom domain focuses on connectivity troubleshooting, plan modifications, and service activation workflows. The Retail domain includes order processing, return handling, and inventory queries. The Airline domain emphasizes booking modifications and policy-compliant rescheduling scenarios.

**Interaction Settings**  Two interaction modes are defined. In Easy mode, a human proxy (implemented via GPT-4.1) provides detailed guidance to the agent. The knowledge base is built exclusively from Easy mode trajectories, ensuring high-quality demonstrations for learning. In Hard mode, human intervention is minimized. The knowledge base combines both Easy and Hard trajectories, testing the agent's robustness to underspecified or noisy instructions.

**Evaluation Criteria**  Task success is measured using domain-specific verification procedures. These include database state checks to validate final outcomes, status checks for confirming service or connection state, natural language verification to ensure correct confirmation statements appear in dialogue, and action matching to confirm that all required steps are completed. Each domain uses a tailored subset of these checks (e.g., Telecom relies primarily on status checks).

| Domain | Test | Train |
|---|---|---|
| Calc | 45 | 2 |
| Chrome | 44 | 2 |
| Writer | 21 | 2 |
| Gimp | 24 | 2 |
| Impress | 45 | 2 |
| Os | 22 | 2 |
| Thunderbird | 13 | 2 |
| Multi-apps | 99 | 2 |
| VLC | 15 | 2 |
| VSCode | 21 | 2 |
| **Total** | **349** | **20** |

Table 2: Test and Train samples across different domains in OSWorld.

| Domain | Test | Train |
|---|---|---|
| Telecom | 105 | 7 |
| Retail | 105 | 7 |
| Airline | 44 | 6 |
| **Total** | **254** | **20** |

Table 3: Task-wise breakdown for $\tau^2$-Bench with assumed 2-shot training samples per domain.

**Domain Characteristics**

- **Telecom:** Connectivity issues, plan management, service activation
- **Retail:** Order processing, returns, inventory queries
- **Airline:** Booking modifications, policy-compliant rescheduling

**Evaluation Criteria**  Task success determined by:

- **Database Checks:** Final state verification

- **Status Checks:** Service/connection state validation
- **NL Checks:** Confirmation statements in dialogue
- **Action Matching:** Required action sequence completion

Note: Each domain uses specific check combinations (e.g., Telecom uses only status checks).

### A.4.3 SPREADSHEETBENCH: REAL-WORLD SPREADSHEET MANIPULATION

**Dataset Overview**  SpreadsheetBench (Ma et al., 2024) consists of 912 instructions collected from four major Excel forums and blogs. Each instruction is paired with spreadsheets reflecting authentic, complex user scenarios, often containing multiple tables and non-standard relational structures. The dataset totals 2,729 test cases, averaging three per instruction. A breakdown of cell-level and sheet-level manipulations is shown in Table 4.

**Task Settings**  The benchmark defines two dimensions of evaluation:

- **Granularity:** Instructions involve either *cell-level* manipulations (specific ranges such as D2:D6) or *sheet-level* manipulations (entire tables or multi-sheet updates).
- **Evaluation:** Performance is measured using an Online Judge (OJ)-style protocol. The *soft* setting (IOI-style) awards partial credit when only some test cases are solved, while the *hard* setting (ICPC-style) requires solutions to succeed on all test cases.

**Agent Configuration**  We evaluate textto4-mini using a function-calling agent connected to a single Python execution tool. The agent translates natural language instructions into Python code for spreadsheet manipulation (e.g., modifying cells, applying formulas, restructuring tables). After each tool call, all formulas in the spreadsheet are recalculated to ensure consistency before proceeding to the next step. This setup provides a controlled environment to assess reasoning, code generation, and execution robustness across diverse spreadsheet tasks.

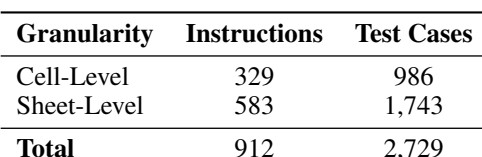

| Granularity | Instructions | Test Cases |
|---|---|---|
| Cell-Level | 329 | 986 |
| Sheet-Level | 583 | 1,743 |
| **Total** | 912 | 2,729 |

Table 4: Cell-level vs. sheet-level distribution in SpreadsheetBench.

### A.5 KB CONSTRUCTION AND RETRIEVAL DETAILS

**Training Data Collection**

- **OSWorld:** 20 successful trajectories (2 per application domain) and 10 for evals.
- $\tau^2$**-Bench:** 20 trajectories balanced across domains and difficulty settings and 10 for evals.
- **SpreadsheetBench:** Uniformly sample 30 trajectories for training and 10 for evaluation.

All numbers are reported on the remaining train set.

**Retrieval Strategy**

- **Query Formation:** For each task we take in the seed Natural Language query as the retrieval query.
- **Retrieval Count:** We take top-3 documents for all the retrieval steps
- **Integration Point:** For SPREADSHEET ENCH and **OSWorld** we insert retrievals in the system prompt augmentation. For $\tau^2$-bench we add perfrom retrieval after each user interaction.

| | Baseline | max_width=3, max_depth=3 | max_width=3, max_depth=10 | max_width=10, max_depth=3 |
|---|---|---|---|---|
| OSworld | 44.20 | 47.56 | 43.83 | 49.32 |

Table 5: OSworld difference in MCTS parameters

# B  QUALITATIVE ANALYSIS

**Exploration on MCTS parameters**  WE evaluate OSworld on two different MCTS parameters.

- Increased Depth: To increase the depth we keep maximum width of the tree as 3 and depth as 10 with max number of iterations as 25. We observe that the Knowledge base over optimizes on the train set leading to a poorer performance on test set.

- Increased Width: For increased width we reverse the parameters where depth is 3 and maximum width is 10 with max iterations 25. We observe many different styles of KBs are generated storing very similar information, these different styles lead to a varied performance on both eval and test set notifying the importance of state search.

We report the numbers on table **??**

# C  EXEMPLAR KNOWLEDGE BASES

## C.1  KNOWLEDGE BASE LEARNED FOR OSWORLD

We showcase a small part of knowledge base learned thought BREW . This demonstrate 3 major parts on which each document is aggregated. These parts discuss when to use a piece of information, why to use the information, how to use the information/tool.

```
## Search and Open Files

**When to use**: Locating documents, spreadsheets, images, or
    downloads for editing, conversion, or attachment.

### How to Perform
- Open **File Manager (Nautilus)** from launcher or system dock
- Press `Ctrl + F` or click the search icon
- Enter part of filename, full name, or wildcard (`*.pdf`, `report*`)
- Use right-click →**Open With** to choose the desired application
- Use the sidebar to navigate to **Downloads**, **Documents**, or
    custom folders

### Additional Actions
- Right-click →**Properties** to check modification date or file type
- Sort results by Date, Type, or Name from the top-right dropdown
- Use `F2` to rename files inline

### Example
- Task: "Edit the file titled `sales_report_march.ods`"
  - Search for `sales` in File Manager
  - Confirm `.ods` type and open with LibreOffice Calc

...

## Insert Images

**When to use**: Adding visual elements to documents, presentations,
    emails, or templates.

### How to Perform
- Navigate to **Insert →Image →From File** (in Writer, Impress,
    Thunderbird)
```

```
1026   - Select an image file (`.png`, `.jpg`, `.svg`) from the file dialog
1027   - Use drag handles to resize; right-click →**Wrap** or **Alignment**
1028       for layout
1029
1030   ### Additional Actions
1031   - In GIMP: **File →Open as Layers** to insert image as a new layer
1032   - Use drag-and-drop from file manager into open document windows
1033   - Use **Format →Image** to apply borders, shadows, or color
1034       corrections (in Writer/Impress)
1035
1036   ### Example
1037   - Task: "Insert the logo.png image into the title slide"
1038    - Open `.odp` file in Impress →Go to Slide 1 →Insert →Image →
1039        Select `logo.png`
1040
1041   ...
1042
1043   ## Export as PDF
1044
1045   **When to use**: Required submission format
1046
1047   ### How to Perform
1048   - Go to **File →Export As PDF**
1049   - Choose output folder (usually **Documents** or **Downloads**)
1050   - Click **Save**, then confirm the exported file opens correctly
1051
1052   ### Additional Actions
1053   - In GIMP or Impress: choose **File →Export As**, then select `.pdf`
1054       from format list
1055   - Use **Save As** to preserve both editable and exported versions
1056       separately
1057
1058   ### Example
1059   - Task: "Export the flyer.xcf as a PDF"
1060    - Open in GIMP →File →Export As →Rename to `flyer.pdf` →Click
1061        Export
```

## C.2 BREW KNOWLEDGE BASE FOR $\tau^2$-BENCH

BREW enable use to learn relevant information for tau bench for across the domains in a single knowledge base. This knowledge base is helpful to use relevant actions from the action pool.

```
1064   ### Additional Actions
1065
1066   * Inform the user:
1067    - Refunds via gift card = immediate.
1068    - Refunds via other methods = -57 business days.
1069
1070   ### Example
1071
1072   * Task: "Cancel a T-shirt order placed yesterday"
1073    * Validate: Status is `pending`
1074    * Reason: "no longer needed"
1075    * Confirm
1076    * Execute tool call
1077
1078
1079   # Exchange Delivered Order
1080
1081   **When to use**:
1082   User wants to swap delivered items for a different variant (e.g., size
1083        or color).
```

**Why to use it**:
To fix sizing or option errors without needing a new purchase.

### How to Perform
- Authenticate user
- Confirm order status is `delivered`
- Get full list of exchange items
> "Please ensure all items for exchange are listed. This step 'cant be
    repeated."
- Ask for refund/payment method
- Confirm:
 > "'Youre exchanging item X for same product, different option.
    Proceed?"
- On confirmation:
 ```python
 request_exchange(order_id="45678", item_exchanges=[...],
    payment_method="paypal")
 ```

### Additional Actions

* Mention: An email will be sent with return instructions
* Validate that the new variant is from the same product

### Example

* Task: "Exchange red shirt for blue in Order #45678"
 * Confirm all exchange items
 * Confirm payment method for difference
 * Execute tool call

### Example

* Task: "Show me my last 2 orders"
 * Authenticate
 * Retrieve and present info

# Deny Unsupported Request

**When to use**:
User asks for an unsupported action (e.g., cancel processed order,
    exchange to different product type, help another user).

**Why to use it**:
To stay compliant with platform policy.

### How to Perform
- Politely reject:
 > "'Im sorry, but I 'cant process that request. 'Its outside the
    allowed scope."

### Example

* Task: "Cancel a processed order"
 * Respond with denial message
# Transfer to Human Agent

**When to use**:
User needs help outside the 'assistants permitted capabilities.

**Why to use it**:
To ensure user gets the right help from trained staff.

### How to Perform

```
- Make tool call:
  ```python
  transfer_to_human_agents()
  ```
- Then inform user:
  > "YOU ARE BEING TRANSFERRED TO A HUMAN AGENT. PLEASE HOLD ON."

### Example

* Task: "Delete a task"
  * Deny deletion
  * Transfer to human
```

## C.3  BREW KNOWLEDGE BASE FOR SPREADSHEETBENCH

```
   Header Extraction
1. Detecting Header Rows
Overview:
To accurately identify header rows, scan the initial region of your
    dataset. This process is crucial for mapping column information
    for further processing.

Approaches:
- Heuristic Checks:
- Look for rows where all cells are strings (e.g., "Name", "Date", "
    Region", "Amount").
- Identify rows with distinctive formatting such as bold text or
    background color.
- Example:
| Name | Date | Region | Amount | |--
    -----|-----------|-----------|-------| | John | 2024-01-01| North
     | 100 |
- Pattern Recognition:
- Use regex to match typical header patterns, such as column names
    starting with uppercase letters.
- Score candidate rows based on the likelihood of being headers.
- Multi-Table Sheets:
- Detect gaps, empty rows, or separators indicating a new table.
- Assign a Table ID to each detected table for later reference.

Edge Cases:
- Merge multi-row headers (e.g., "Sales" over "2024", "2025" becomes "
    Sales 2024", "Sales 2025").
- Fill in missing headers by inferring from context.

2. Assigning and Validating Headers
Overview:
Once headers are detected, assign them programmatically and ensure
    they match expected schema and data types.

Implementation:
- Column Naming:
- Set names in code, e.g., df.columns = ["Name", "Date", "Region", "
    Amount"].
- Schema Mapping:
- Map headers to a standardized schema, using external files or user
    prompts.
- Example:
- Raw header: "Amt"; Mapped header: "Amount"
- Quality Checks:
- Detect duplicate or empty headers ("Date", "Date" becomes "Date_1",
    "Date_2").
- Validate each column's expected data type.
```

3. Automation and Usability Enhancements
Overview:
Enhance usability and automation to streamline header extraction and
    user interaction.

Features:
- Freeze Panes:
- Automatically freeze header rows in Excel for easier navigation.
- Highlighting:
- Use colored formatting to visually distinguish headers.
- Example:
- Yellow fill for header row.
- Documentation:
- Log extraction logic and confidence scores for each detected header.
- Integration:
- Build header extraction into ETL pipelines and record process
    metadata.

Block Detection
1. Identifying Block Boundaries
Overview:
Block detection segments data into logical units or tables.

Methods:
- Boundary Detection:
- Find empty rows, repeated labels, or formatting changes.
- Example:
| Name | Amount | |------|--------| | John | 100 | | | | | <-- Empty row
    indicates new block | Name | Amount | | Alice| 200 |
- Machine Learning:
- Train classifiers to detect block boundaries based on cell patterns.

Advanced:
- Detect nested blocks or hierarchies using indentation or merged
    cells.
- Identify summary blocks with keywords like "Total" or "Summary".

2. Processing and Tracking Blocks
Overview:
Once blocks are detected, assign IDs and enable block-level analysis.

Actions:
- Block ID:
- Assign unique IDs (e.g., Block_001, Block_002).
- Analysis:
- Perform group-by or aggregation within each block.
- Example:
- Sum "Amount" for Block_001: 100 + 150 = 250

3. Additional Block Actions
Overview:
Enable modular analysis and reporting at the block level.

Features:
- Summary Rows:
- Add computed totals/averages for each block.
- Export/Save:
- Save blocks as separate files or sheets.
- Example:
- Export Block_001 to "block1.csv"

Search for Values or Patterns
1. Search Execution Methods
Overview:
Efficiently locate specific values or patterns in your data.

Techniques:
- Manual Tools:
- Use Ctrl + F in Excel for quick lookups.
- Programmatic Search:
- Scan all cells using loops or vectorized code.
- Example:
- Find all instances of "North" in the "Region" column.
- Pattern Matching:
- Support exact, wildcard (*Total*), and regex (\d{4}-\d{2}-\d{2} for dates).

2. Recording and Highlighting Results
Overview:
Log and visualize search matches for user review.

Actions:
- Logging:
- Record coordinates (e.g., Sheet1, Row 3, Col "Region").
- Highlighting:
- Apply conditional formatting to search hits.

3. Advanced Search Scenarios
Overview:
Handle complex or large-scale search requirements.

Scenarios:
- Merged Cells:
- Search within merged cells or across multiple sheets.
- Export:
- Export found results for further analysis.
- Example:
- Export all rows containing "John" to "john_results.csv"

Writeback Results
1. Output Placement
Overview:
Choose where and how to insert results.

Options:
- Target Columns:
- Select existing or blank columns for output.
- Appending:
- Add new columns for flags, counts, or statuses.
- Example:
- Add "Approved_Flag" column next to "Status".

2. Writing and Styling Results
Overview:
Automate and style the output for visibility.

Methods:
- Formulas/Code:
- Use code (e.g., ws.cell(row, col).value = result) to insert results.
- Styling:
- Bold, borders, or colors for output cells.
- Example:
- Green fill for "Success", red for "Error".

3. Audit and Protection
Overview:
Maintain the integrity and traceability of results.

Measures:
- Lock Columns:

```
- Prevent edits to output columns.
- Timestamps/User Info:
- Add audit trail for writebacks.
- Example:
- "2024-06-01, User: admin"

Difference in State
1. Sheet Comparison
Overview:
Identify changes between input and output sheets.

Process:
- Load Sheets:
- Read both sheets into memory.
- Compare Cells:
- Detect differences by position and value.

2. Recording and Reporting Differences
Overview:
Log and report all detected changes.

Actions:
- Log Mismatches:
- Record cell coordinates and values.
- Example:
- Cell B3: "North" →"South"
- Export Diff Report:
- List all detected differences for review.

3. Visualization and Automation
Overview:
Make changes visible and automate validation.

Features:
- Highlight Changes:
- Color code changed cells.
- Automate Checks:
- Integrate diff comparisons into test scripts.

Column Selection
1. Selection Criteria
Overview:
Choose relevant columns for analysis.

Methods:
- Labels/Indices:
- Select by name or position.
- Dynamic Rules:
- E.g., all numeric columns.
- Assign Roles:
- Example: "ID", "Date", "Metric"

2. Preparation and Validation
Overview:
Prepare columns for consistent use.

Actions:
- Rename/Relabel:
- Standardize column names.
- Validate Types:
- Ensure columns are of expected type.
- Example:
- "Date" column as datetime.

3. Reusability
```

```
Overview:
Save and reuse column selections.

Features:
- Presets:
- Save selection profiles.
- Downstream Use:
- Use validated columns in subsequent processes.

Filter Rows
1. Filtering Methods
Overview:
Refine your dataset with filters.

Techniques:
- Spreadsheet Tools:
- Use built-in filters.
- Code Logic:
- Filter with code (e.g., df[df['Status'] == 'Approved']).
- Multiple Criteria:
- Combine conditions (AND/OR).
- Example:
- Status = "Approved" AND Amount > 100

2. Helper Columns and Complex Filters
Overview:
Simplify filtering using helper columns.

Actions:
- Helper Columns:
- Compute intermediate flags.
- Document Logic:
- Record filtering rules for audit.

3. Post-Filter Actions
Overview:
Visualize and export filtered data.

Features:
- Highlighting:
- Grey-out filtered-out rows.
- Export:
- Save the filtered dataset.

Merge Tables
1. Key-Based Merging
Overview:
Combine tables using shared keys.

Techniques:
- Join Operations:
- Use VLOOKUP, JOIN, or code merges.
- Example:
- Merge "Customer_ID" from two tables.
- Align Data:
- Match on columns like "ID", "Name".

2. Stack-Based Merging
Overview:
Append tables when keys 'arent needed.

Methods:
- Vertical Append:
- Combine rows from similar tables.
- Deduplicate:
```

- Remove duplicate records.

3. Tracking and Audit
Overview:
Track source and unmatched records.

Actions:
- Source Column:
- Add "Source" to indicate origin.
- Highlight Unmatched:
- Mark or export mismatched rows.

Pivot or Unpivot
1. Pivoting Data
Overview:
Summarize data using pivots.

Methods:
- PivotTables:
- Group by row/column dimensions.
- Example:
- Sum "Amount" by "Region".
- Aggregation:
- Choose SUM, AVG, COUNT, etc.

2. Unpivoting (Melting) Data
Overview:
Reshape data from wide to long format.

Techniques:
- Melt Operations:
- Convert columns into rows.
- Example:
-
| Year | Sales_2019 | Sales_2020 | |------|------------|------------|
→
| Year | Sales_Year | Value |
- Flexible Restructuring:
- Selectively unpivot non-ID columns.

3. Post-Pivot Actions
Overview:
Prepare pivoted data for export.

Features:
- Flatten Pivot Table:
- Convert back to flat for further analysis.
- Reorder/Rename:
- Clarify pivoted fields.

Map with Lookup Tables
1. Mapping Techniques
Overview:
Standardize data using lookups.

Methods:
- Functions:
- Use VLOOKUP, merge with dictionaries.
- Code-to-Label:
- Example:
- Code "N" →Label "North"

2. Application and Fallbacks
Overview:
Apply lookups and handle missing values.

```
Actions:
- Apply Mappings:
- Across selected columns.
- Handle Missings:
- Use defaults for missing codes.

3. Audit and Display
Overview:
Ensure mapping transparency.

Features:
- Cache Mappings:
- Store for repeated use.
- Display Codes/Labels:
- Show both for clarity.

Fill Missing Data
1. Choosing Fill Methods
Overview:
Impute missing data appropriately.

Techniques:
- Forward/Backward Fill:
- Fill gaps with prior/next value.
- Default Values:
- Use fixed placeholder (e.g., 0, "Unknown").
- Contextual Example:
- Dates: Fill missing month with last known month.

2. Application and Auditing
Overview:
Apply fills and flag for review.

Actions:
- Targeted Filling:
- Apply to specific columns/rows.
- Flag Filled Cells:
- Highlight for later review.

3. Documentation
Overview:
Keep fill logic transparent.

Features:
- Record Logic:
- Document assumptions and methods.
- Audit Trail:
- Track all changes.

Flag Rows or Cells
1. Defining Flag Rules
Overview:
Establish criteria for flagging.

Examples:
- Simple Rule:
- Flag where Amount < 0
- Complex Rule:
- Flag where Status = "Pending" and Amount > 1000

2. Applying Flags
Overview:
Insert flags and summarize.
```

Actions:
- Flag Column:
- Add "Flag" column with "Yes"/"No".
- Export Flagged Rows:
- Save for further inspection.

3. Advanced Flagging
Overview:
Use multiple criteria and document.

Features:
- Multi-Criteria:
- Combine several rules for granular checks.
- Notes:
- Document flagging rationale.

Sort Data
1. Setting Sort Criteria
Overview:
Organize data for analysis.

Options:
- Sort Columns:
- By value, ascending/descending.
- Multi-Level:
- E.g., sort by "Region", then by "Amount".

2. Applying Sorts
Overview:
Implement sorting programmatically or manually.

Methods:
- Spreadsheet Tools:
- Built-in sort features.
- Code:
- E.g., df.sort_values(['Region', 'Amount'])

3. Post-Sort Actions
Overview:
Finalize sorted data.

Actions:
- Renumber Rows:
- Update indices.
- Highlight Extremes:
- Mark top/bottom values.

Validate Data
1. Validation Checks
Overview:
Ensure data meets required standards.

Checks:
- Type:
- Ensure numeric columns contain numbers.
- Range:
- E.g., "Amount" > 0.
- Pattern:
- Date columns match YYYY-MM-DD.
- Business Rule Example:
- "Start Date" < "End Date"

2. Marking and Reporting
Overview:
Visualize and report errors.

```
Actions:
- Highlight Invalids:
- Color-code errors.
- Export Summary:
- Table of error counts and locations.

3. Integration in Workflow
Overview:
Make validation a routine part of processing.

Features:
- Pre-Processing Step:
- Validate before analysis.
- Automation:
- Integrate into data pipelines.

Split Sheets or Data
1. Defining Split Rules
Overview:
Segment data for modular analysis.

Methods:
- By Category:
- E.g., split by "Region".
- By Date Range:
- E.g., split by year.

2. Exporting Segments
Overview:
Save segments for separate use.

Actions:
- Export Files:
- "North_Region.csv", "South_Region.csv"
- Consistent Formatting:
- Ensure identical columns and styling.

3. Automation and Documentation
Overview:
Automate splitting and track provenance.

Features:
- Automation:
- Use scripts/macros for repeated splits.
- Documentation:
- Record rules and export logs.
```

# D QUALITATIVE ANALYSIS OF BREW-GENERATED KNOWLEDGE BASES

This section presents a comprehensive qualitative analysis of knowledge bases generated through the BREW technique applied to two distinct agent training environments: OSWorld and $\tau^2$Bench described in the section before. The analysis examines knowledge representation patterns, procedural sophistication, and domain-specific learning characteristics extracted from CUA agent behaviors, providing insights into the effectiveness and scope of knowledge distillation techniques across diverse task environments.

## D.1   CROSS-DOMAIN KNOWLEDGE BASE ANALYSIS

### D.1.1   BASE STRUCTURE & ORGANIZATION

**Schema Consistency and Evolution**: Both knowledge bases demonstrate consistent structural schemas, though adapted to their respective domains. The OSWorld KB employs a four-part schema (contextual triggers, procedural steps, extended capabilities, concrete instantiation), while the $\tau^2$Bench KB extends this to a five-part structure, adding explicit purpose rationale ("Why to use it"). This evolution suggests that BREW adapts its extraction patterns to domain-specific requirements— conversational commerce demands explicit justification for actions due to customer interaction contexts.

**Taxonomic Organization Principles**: The OSWorld KB reveals a **capability-based taxonomy** organized around computational tasks: file operations, document processing, inter-application workflows, and data visualization. Each category represents a distinct computational domain with specific tool requirements and interaction patterns. In contrast, the $\tau^2$Bench KB employs a **lifecycle-based taxonomy** structured around transactional states: order creation, modification, fulfillment, and post-delivery operations. This organizational difference reflects fundamental domain characteristics— desktop automation focuses on tool orchestration, while conversational commerce centers on process management.

**Hierarchical Task Decomposition**: Both KBs demonstrate sophisticated hierarchical reasoning, but through different decomposition strategies. OSWorld exhibits **technical decomposition**, breaking complex operations like "Create Charts from Data" into constituent technical steps (data selection, chart insertion, customization, formatting). $\tau^2$Bench shows **process decomposition**, structuring operations like order modification into authentication, validation, confirmation, and execution phases. This suggests BREW successfully identifies domain-appropriate decomposition strategies rather than applying uniform patterns.

**Knowledge Boundary Definition**: Both KBs explicitly encode operational boundaries, but through contrasting mechanisms. OSWorld boundaries are **capability-constrained**—determined by available applications and system resources. $\tau^2$Bench boundaries are **policy-constrained**—explicitly defined through "Deny Unsupported Request" patterns and escalation protocols. This difference highlights how knowledge extraction adapts to domain-specific constraint types.

### D.1.2   PROCEDURAL KNOWLEDGE GROUNDING

**Context-Dependent Action Selection**: Both domains demonstrate sophisticated context awareness, but grounded in different environmental factors. OSWorld exhibits **application-context sensitivity**, where identical operations (e.g., image insertion) require different procedures across LibreOffice Writer, Impress, GIMP, and Thunderbird. The agent learned application-specific affordances and interaction patterns rather than generic command sequences. $\tau^2$Bench demonstrates **state-context sensitivity**, where available actions depend on order status (pending vs. delivered), payment methods, and authentication levels. This reveals learned understanding of business process constraints and temporal operation windows.

**Error Prevention and Validation Workflows**: Both KBs incorporate sophisticated error prevention mechanisms, but grounded in domain-specific failure modes. OSWorld emphasizes **technical validation**: file integrity checks ("confirm the exported file opens correctly"), application state verification, and multi-step confirmation for irreversible operations. $\tau^2$Bench emphasizes **transactional validation**: authentication cascades, confirmation dialogues with standardized templates, and explicit user consent protocols. The emergence of defensive programming practices across both domains suggests these represent fundamental principles of reliable agent behavior.

**State-Dependent Decision Logic**: The procedural knowledge in both domains demonstrates sophisticated state machine reasoning. OSWorld exhibits **application state awareness**—understanding when applications are ready for input, when files are loaded, and when operations can be safely executed. Window management and application switching reveal learned understanding of desktop metaphors and resource constraints. $\tau^2$Bench demonstrates **business process state awareness**—finite state machine reasoning where order lifecycle states determine available operations. The agent learned that pending orders enable modification while delivered orders unlock return workflows, indicating internalized understanding of business logic constraints.

**Security and Authentication Grounding**: While OSWorld operates in a trusted desktop environment with minimal explicit security concerns, $\tau^2$Bench reveals pervasive **authentication-first paradigms**. Nearly every transactional operation begins with identity verification through email, name, and zip code combinations. The KB demonstrates **graduated security reasoning**: information retrieval requires basic authentication while financial transactions trigger rigorous verification protocols. This contrast highlights how procedural knowledge adapts to domain-specific security requirements.

**Cross-Application vs. Cross-Process Orchestration**: OSWorld demonstrates **technical orchestration**—coordinating multiple applications (Chrome, LibreOffice suite, File Manager, GIMP) to accomplish complex workflows. The "Navigate Between Applications" section reveals learned behaviors for window management, application switching, and resource coordination. $\tau^2$Bench exhibits **process orchestration**—coordinating authentication, validation, confirmation, and execution phases across different operational contexts. Both forms of orchestration require sophisticated temporal reasoning and constraint management, but applied to different environmental complexity types.

**Failure Mode Internalization**: Both KBs reveal learned understanding of domain-specific failure modes. OSWorld incorporates file validation, application crash recovery suggestions, and verification steps for critical operations. $\tau^2$Bench includes explicit escalation protocols ("Transfer to Human Agent"), policy compliance mechanisms, and irreversibility warnings for financial operations. The consistent emergence of failure-aware procedures suggests that agents successfully internalize risk assessment and mitigation strategies during training.

**Domain-Specific Communication Patterns**: The procedural knowledge reveals distinct communication paradigms appropriate to each domain. OSWorld procedures are **task-oriented** with minimal user interaction—focusing on efficient command execution and verification. $\tau^2$Bench procedures are **dialogue-oriented** with standardized customer interaction templates, confirmation protocols, and expectation management communications. This adaptation demonstrates that BREW extracts not just procedural logic but domain-appropriate interaction modalities.

The cross-domain analysis reveals that BREW successfully extracts procedural knowledge that is both **structurally consistent** (following learnable organizational patterns) and **contextually grounded** (adapted to domain-specific constraints, failure modes, and interaction requirements). This dual capability suggests significant potential for knowledge transfer across related domains while maintaining appropriate domain-specific adaptations.

