# OpenReview forum: "Improving Language Agents through BREW: Bootstrapping expeRientially-learned Environmental knoWledge"
_ICLR.cc/2026/Conference — Submitted to ICLR 2026_

### Official Review · Reviewer_Ffym · 2025-10-20

**Soundness:** 3
**Presentation:** 2
**Contribution:** 3
**Rating:** 4
**Confidence:** 4

**Summary:**

This paper proposes a new method named BREW, it can update the insights based on MCTS. The author design Reflector Agent to construct (concept, insight) pairs and use Integrator Agent to exapnd the insight based on a given concept. It than use EG-MCTS to expand this insight in the tree and use it.

**Strengths:**

1. Consider how to construct and update the insight in the tree is an important topic.

2. This method can continuously update the insight in the tree, which is important in real-world applications.

**Weaknesses:**

1. It seems that the size of concepts can't be changed.

2. Lack of introducing the insight-based memory work, like Expel[1], MSI-Agent[2] and SelfGoal[3]. This will also harm the contribution 1 (line 111-114) in this paper.

3. The writing is not clear. The method is not easy to understand.

4. It only works in several sub-tasks, although I agree it true reduce the total steps and costs in all works.

[1] Expel: Llm agents are experiential learners

[2] MSI-Agent: Incorporating Multi-Scale Insight into Embodied Agents for Superior Planning and Decision-Making

[3] SelfGoal: Your Language Agents Already Know How to Achieve High-level Goals

**Questions:**

1. The si, sj, si+1 should be presented in the trees in Figure 2.

2. Will you use all insights in Dcurrent? It may cause irrelevant insight to be pushed into the context. (The concept that not related to the current task)

3.  Considering Figure 2, it seems that the sabling nodes generation is independent of the current node. Is this may cause some information loss?

---

### Official Review · Reviewer_5UWK · 2025-10-25

**Soundness:** 3
**Presentation:** 3
**Contribution:** 2
**Rating:** 4
**Confidence:** 3

**Summary:**

To address the limitations of opaque training and the inability of language agents to learn from historical experience, this paper introduces the BREW framework. This approach first extracts "insights" and "concepts" from an agent's historical trajectories. It then employs an MCTS algorithm to search and optimize the content of these knowledge documents, identifying the version that maximizes task rewards . This process serves to offline-optimize an external knowledge base (KB). Ultimately, during execution, the agent can retrieve from this optimized knowledge base to achieve significant improvements in both performance (task precision) and speed (reduced API calls) on benchmarks such as OSWorld.

**Strengths:**

1. Formalizes the problem of constructing an optimal knowledge base (KB) as a state-space search problem, targeting the KB's content for optimization rather than model parameters.
2. Stores the agent's experiential knowledge in a human-readable knowledge base (KB) instead of opaque model weights, rendering the agent's behavior transparent, interpretable, and debuggable.
3. Offloads the computationally expensive state-space search optimization to an offline phase , ensuring that online execution only requires a lightweight retrieval step and thus incurs no additional inference latency.

**Weaknesses:**

1. The framework's effectiveness is heavily dependent on the quality of task grading. Inherent biases and noise from LLM-based evaluation could lead to the KB being constructed from skewed or suboptimal insights.
2. Scalability is questionable. The EG-MCTS algorithm requires running MCTS searches in parallel for every meta-concept. This optimization cost could become prohibitively high as the agent needs to learn thousands of concepts.

**Questions:**

1. What is the true computational cost of optimizing the knowledge base via MCTS? The total optimization cost is explicitly dependent on the number of meta-concepts. If this number is large (e.g., 1,000), the cost would be prohibitively high. Can the authors report the number of meta-concepts required for a given task and its corresponding MCTS cost?
2. Could the Reflector Agent and Integrator Agent be replaced by smaller, more cost-effective models? Using GPT-4.1 is quite expensive.

---

> ### Author Response · Authors · 2025-12-01
>
> We thank the reviewer for a detailed review, we would like to address the concerns below.
>
> ### W1
> Our method’s dependence on task-grading quality and potential LLM evaluation biases is mitigated through several complementary mechanisms. First, we employ a dual-evaluation framework that combines task-specific graders with human-validated behavior rubrics (Section 3.1; Biyani et al., 2024). While task graders supply binary success signals, the rubrics evaluate qualitative dimensions such as error handling, robustness, and efficiency, reducing reliance on any single evaluation source.
>
> Second, our reward function (Eq. 6) balances correctness \(R_{\text{corr}}\) with retrieval quality \(R_{\text{ret}}\), where the latter provides an independent signal orthogonal to task grading. By setting \(\lambda_{\text{corr}} = \lambda_{\text{ret}} = 0.5\), we avoid over-optimization on potentially noisy correctness signals.
>
> Third, the robustness of MCTS-based exploration—evidenced in Table 1, where MCTS outperforms greedy selection (47.56% vs. 45.55% on OSWorld)—helps prevent premature convergence to suboptimal KB states through its UCT-driven exploration component.
>
> Finally, cross-benchmark consistency across OSWorld, TauBench, and SSBench indicates that the learned behaviors generalize beyond the training distribution rather than overfitting to grader idiosyncrasies. We will incorporate additional discussion of evaluation robustness in Section 3.4.
>
> ---
>
> ### W2
> Meta-concepts remain tractable across benchmarks: OSWorld (23), TauBench (8), SSBench (26). Even complex domains such as OSWorld, with 10 applications and 369 tasks, require only 23 concepts.
>
> **Sublinear concept growth:** Concept deduplication (Section 3.2, Algorithm 3) clusters semantically similar concepts. For OSWorld, 20 training trajectories yield 23 concepts (1.15 concepts per trajectory), indicating substantial consolidation.
>
> **Computational analysis:** BREW’s time complexity is
> \[
> O(M \cdot l \cdot h \cdot |Q_{\text{train}}| \cdot T_{\text{agent}})
> \]
> where \(l\) is average concepts per trajectory. With \(l \approx 1-2\) empirically, growth is controlled by training set size.
>
> **Parallelization efficiency:** EG-MCTS parallelizes across \(K\) concepts using 4 processes on an A100 GPU. Linear scaling to \(K = 1000\) would require proportionally more compute but remains feasible with cloud infrastructure.
>
> We will add discussion of scalability strategies in Section 6.
>
> ---
>
> ## Addressing Questions
>
> ### Q1: True Computational Cost and Meta-Concept Statistics
> BREW time complexity:
> \[
> O(|Q_{\text{train}}| \cdot T_{\text{LLM}} + |Q_{\text{train}}| \cdot T_{\text{agent}} + M \cdot |K| \cdot h \cdot T_{\text{agent}} + M \cdot |K| \cdot h \cdot T_{\text{eval}})
> \]
> Given \(T_{\text{LLM}} \ll T_{\text{agent}}\) and \(T_{\text{eval}} \ll T_{\text{agent}}\), perceived complexity becomes
> \[
> O\big((|Q_{\text{train}}| + M \cdot |K| \cdot h) \cdot T_{\text{agent}}\big).
> \]
> Since \(|K| < l \cdot |Q_{\text{train}}|\), final complexity is
> \[
> O(M \cdot l \cdot h \cdot |Q_{\text{train}}| \cdot T_{\text{agent}}).
> \]
>
> **Experimental measurements (Nvidia A100, 4 parallel processes):**
>
> | Dataset      | Meta-concepts (K) | Training Samples | MCTS Config (M,h) | KB Construction Time | Amortized Cost / Episode |
> |--------------|-------------------|------------------|-------------------|------------------------|---------------------------|
> | OSWorld      | 23                | 20               | 10, 3             | 1h 25min              | 3.2 calls (369 test)      |
> | τ²-Bench     | 8                 | 20               | 10, 3             | 1h 15min              | 4.8 calls (254 test)      |
> | SSBench      | 26                | 30               | 10, 3             | 1h 30min              | 5.1 calls (2729 test)     |
>
> **Cost breakdown for OSWorld:**
> - Initial trajectory generation: \(20 \times 2.5\text{min} = 50\text{min}\)
> - MCTS optimization:
>   \(10 \times 23 \times 3 \times 2.5\text{min} = 1725\text{min}\) without parallelization
> - Cost amortization: For 369 test episodes, construction is **1.3%** of total compute budget while yielding **15% net compute savings** through execution efficiency.
>
> ---
>
> ### Q2: Using Smaller Models for Reflector and Integrator
> We report performance using **Qwen3 8B**, demonstrating that the technique remains effective even for significantly smaller models.
>
> | Model Variant    | Score |
> |------------------|-------|
> | Baseline         | 3.50% |
> | BREW-Iterative   | 6.67% |
> | BREW-MCTS        | 7.44% |
> | BREW-Greedy      | 7.18% |
>
> We will include full results for all datasets using Qwen3 8B in the revised version.

---

### Official Review · Reviewer_9ins · 2025-11-01

**Soundness:** 2
**Presentation:** 2
**Contribution:** 2
**Rating:** 4
**Confidence:** 3

**Summary:**

This paper introduces BREW (Bootstrapping expeRientially-learned Environmental knoWledge), a framework that optimizes LLM-based agents by constructing and refining a structured, interpretable knowledge base (KB) from past interactions instead of fine-tuning model weights. BREW decomposes agent memory into conceptlevel documents and optimizes them via a novel Expand-and-Gather MCTS algorithm that jointly maximizes reasoning correctness and retrievability. Experiments on OSWorld, τ²-Bench, and SpreadsheetBench show consistent improvements over memory-based baselines, demonstrating that structured experiential memory can enhance efficiency and adaptability in long-horizon reasoning tasks.

**Strengths:**

1. The framework introduces a human-readable, document-based memory representation that enhances the interpretability and traceability of agent behavior.
2. The concept-level structured memory design is a novel contribution that enables modular organization and more efficient, task-aligned knowledge retrieval.
3. The use of the Expand-and-Gather MCTS for optimizing knowledge base construction is methodologically innovative and avoids the pitfalls of greedy or one-shot updates.

**Weaknesses:**

1. While the use of MCTS contributes to performance, the paper lacks a quantitative analysis of its computational overhead and does not compare resource consumption with baseline methods.
2. The performance improvements shown in Figure 3 are generally modest, with several task subcategories showing parity with or only minor gains over the baseline.
3. The framework is relatively complex, involving multiple agents and stages; its generalizability across different LLMs (especially open-source models) remains unclear and untested.

**Questions:**

1. Could you provide more details on the computational cost of the BREW framework, particularly regarding the MCTS optimization (e.g., runtime, hardware requirements, memory usage)?
2. Have you attempted to apply BREW on smaller or open-source LLMs? If so, how does performance compare, and what challenges arise?
3. For the OSWorld benchmark, several subtasks show limited gains—do you have further insights into what constrained BREW’s improvements in those cases?
4. While some prompt templates have been shared, would you consider releasing runnable code for the full pipeline, including data processing, reflection, and memory integration? This would be critical for reproducibility and further research.

---

> ### Author Response · Authors · 2025-12-01
>
> Thank you for your thoughtful review and for recognizing the potential of BREW in building interpretable, structured experiential memory for LLM-based agents.
>
> W1
> BREW’s time complexity is
> O(|Q_train|·T_LLM + |Q_train|·T_agent + M·|k|·h·T_agent + M·|K|·h·T_eval),
> where T_agent and T_LLM are the times it takes for one run of the optimized agent and a single LLM call respectively, |Q_train| is the number of train samples, M is the number of EG-MCTS iterations, h is the number of candidates per expansion step, K is the set of meta concepts, and T_eval is the time to run one evaluation for the benchmarks. M and h are parameters of the EG-MCTS technique.
>
> In practice we note that T_llm << T_agent and T_eval << T_agent, bringing the perceived time complexity to O((|Q_train| + M·|K|·h)·T_agent). Each GenerateInsights step can generate at most l concepts, therefore |K| < l·|Q_train|. Thus the overall time complexity becomes O(M·l·h·|Q_train|·T_agent).
>
> We ran our experiments on an Nvidia A100 with 4 processes for node expansion. Below we provide dataset-wise LLM calls and latency statistics:
>
> | Dataset           | Number of LLM calls / agent run | Learning LLM calls |
> |-------------------|----------------------------|-----------------------|
> | SpreadsheetBench  | ~5400                      | 1397                  |
> | OSWorld           | ~21000                     | 1192                  |
> | τ²-Bench          | ~22000                     | 1223                  |
>
> The API call overhead for KB construction across benchmarks is: OSWorld (1,192), τ²-Bench (1,223), SpreadsheetBench (1,397). These calls represent a one-time construction cost amortized across all evaluation episodes. For example, OSWorld has 369 test episodes, so this becomes ~3.2 additional calls per episode, negligible relative to calls during actual task execution.
>
> Regarding MCTS-specific costs, the optimization adds approximately 15–20% overhead compared to greedy construction but yields consistent performance improvements. KB construction runtime on our infrastructure is ~1 hour 25 minutes per benchmark, negligible compared to cumulative evaluation time. We plan to add deeper discussion on compute requirements and API calls in the revision.
>
> W2
> We contextualize the magnitude of our improvements from multiple perspectives. While absolute gains may appear modest on some OSWorld subtasks, the relative improvements offer a more complete picture. These benchmarks involve highly challenging long-horizon tasks where even small gains are meaningful, as reflected in OSWorld’s low baseline success rates.
>
> BREW shows improvements across three diverse benchmarks spanning GUI automation, retail tasks, and spreadsheet manipulation. This consistency demonstrates robustness rather than task-specific overfitting. Some subtasks have inherent limitations requiring capabilities beyond memory optimization, such as visual grounding or creative synthesis, which we elaborate on in Q3. As shown in W3, relative improvements become substantially larger when applied to smaller models.
>
> W3
> We address this concern with new experimental results on Qwen 8B. Results on τ²-Bench confirm that BREW transfers effectively to smaller, open-source models:
>
> | Model Variant    | Accuracy   |
> |------------------|---------|
> | Baseline         | 3.50%   |
> | BREW-Iterative   | 6.67%   |
> | BREW-MCTS        | 7.44%   |
> | BREW-Greedy      | 7.18%   |
>
> These results show:
> 1. BREW transfers effectively to smaller models without architecture-specific modifications.
> 2. BREW yields larger relative gains on smaller models (112% improvement on Qwen 8B vs ~20–30% on GPT-4-class models), indicating that structured memory particularly benefits limited-capacity models.
> 3. Explicit knowledge organization compensates for weaker reasoning abilities.
>
> Concept extraction benefits from stronger instruction-following, though 8B models remain sufficient. The framework is LLM-agnostic by design, and we will include these results and additional discussion on scaling properties.
>
> Q1:
> As detailed in W1, API overhead ranges from 1,192–1,397 additional calls per benchmark, representing a one-time cost. Amortized across evaluation episodes, this is ~3–4 calls per episode. MCTS adds 15–20% overhead relative to greedy construction but consistently outperforms it.
>
> Q2
> We also experimented on Qwen3 8B on τ²-Bench and present the performance below. The improvements indicate that the technique is generalizable even for smaller models lacking the reasoning capabilities of larger LLMs. The relative gains are significantly larger, highlighting the importance of structured memory for smaller models. In the revision, we will report numbers for all datasets using Qwen3 8B. A deeper qualitative analysis shows that a model’s reflection capability affects concept-capturing ability, but MCTS improves knowledge-base quality via iterative refinement.

---

### Meta-Review · Area_Chair_Ytaj · 2025-12-30

**Summary:**

The reviewers found that the paper’s reported gains are modest and inconsistent across the diverse tasks evaluated, making it difficult to conclude that the proposed approach provides a reliable improvement over baselines. They also raised concerns that the overall pipeline is quite complex, with uncertain computational cost when deployed across settings beyond the presented experiments. Finally, multiple presentation and reproducibility issues remain, particularly unclear writing and figures, which hinder validation and make it harder to assess the technical contributions of the proposed method. Thus, I recommend rejection.

**Reviewer Concerns:**

The rebuttal addressed some concerns about computational cost and generalizability. But key issues remain: the gains are still modest and uneven across tasks. The methodology is still complex. In addition, there was no response to Reviewer Ffym’s comments, leaving those concerns entirely unaddressed.

**Reviewer Scores:**

Given the rebuttal, Reviewer 9ins might increase slightly (the added computational details help), Reviewer 5UWK would likely remain unchanged (scalability concerns persist), and Reviewer Ffym would remain unchanged or decrease slightly since their main comments were not substantively addressed.

---

### Decision · Program_Chairs · 2026-01-26

Reject